# Single-cell TCR sequencing reveals phenotypically diverse clonally expanded cells harboring inducible HIV proviruses during ART

Pierre Gantner[1], Amélie Pagliuzza[2], Marion Pardons[1], Moti Ramgopal[3], Jean-Pierre Routy [4], Rémi Fromentin [2] & Nicolas Chomont [1,2✉]

Clonal expansions occur in the persistent HIV reservoir as shown by the duplication of proviral integration sites. However, the source of the proliferation of HIV-infected cells remains unclear. Here, we analyze the TCR repertoire of single HIV-infected cells harboring translation-competent proviruses in longitudinal samples from eight individuals on anti-retroviral therapy (ART). When compared to uninfected cells, the TCR repertoire of reservoir cells is heavily biased: expanded clonotypes are present in all individuals, account for the majority of reservoir cells and are often maintained over time on ART. Infected T cell clones are detected at low frequencies in the long-lived central memory compartment and over-represented in the most differentiated memory subsets. Our results indicate that clonal expansions highly contribute to the persistence of the HIV reservoir and suggest that reservoir cells displaying a differentiated phenotype are the progeny of infected central memory cells undergoing antigen-driven clonal expansion during ART.

[1] Department of Microbiology, Infectiology and Immunology, Université de Montréal, Montreal, QC, Canada. [2] Centre de Recherche du Centre Hospitalier de l'Université de Montréal, Montreal, QC, Canada. [3] Midway Immunology & Research Center, Fort Pierce, FL, USA. [4] Division of Hematology & Chronic Viral Illness Service, McGill University Heath Centre, Montreal, QC, Canada. ✉email: nicolas.chomont@umontreal.ca

 **1**

The persistence of replication competent proviruses in memory CD4+ T cells is the main barrier to viral eradication in people living with HIV[1–3]. Latently infected cells are maintained during effective ART through both cell survival and cell division signals, promoting clonal expansions of HIV-infected cells[4,5], as demonstrated by the duplication of integration sites[6–9] and/or HIV genomes[10–18]. During ART, clonally expanded HIV-infected cells have the ability to expand and contract over time[10,15,19–21]. Several mechanisms are thought to contribute to the dynamics of the HIV reservoir[22], including (1) antigen driven proliferation[23], (2) homeostatic proliferation[4,24,25], and (3) viral genome integration into specific cellular genes that may promote cell proliferation[6,8,9]. Altogether, these studies indicate that the reservoir is highly dynamic during ART but the relative contributions of these mechanisms remain unclear.

CD4+ T cells harboring replication-competent genomes are phenotypically diverse with multiple memory and functional CD4+ T cells subsets contributing to HIV persistence[4,14,26,27]. Combining flow cytometry cell sorting and near full length HIV DNA sequencing revealed that intact (and potentially replication competent) viral genomes are enriched in specific CD4+ T cell subsets, such as Th1 cells and effector memory cells[14,28]. However, none of these approaches allowed to simultaneously investigate the inducibility, location and dynamics of individual proviruses over time in virally suppressed individuals.

Here, we take advantage of the uniqueness of the T-cell receptor (TCR) within a given T-cell clone[29–31] to unravel the phenotype and dynamics of the inducible HIV reservoir during ART. We hypothesize that duplication of TCR clonotypes within the pool of HIV-infected cells will reflect the dynamics of clonal expansion as well as the persistence of individual HIV-infected clones[32].

## Results

### TCRβ sequencing and phenotyping of single HIV-infected cells.

We developed a novel approach, using the unique VDJ rearranged sequence of TCRβ as a cellular tag, to track individual HIV-infected cells (Fig. 1a). CD4+ T cells isolated from the blood of virally suppressed individuals were stimulated for 24 h with PMA/ionomycin to induce p24 expression, thus unraveling the translation-competent HIV reservoir. Clonotypic characterization of individual HIV-infected cells was performed by combining single-cell sorting of HIV-infected (p24+) cells by HIV-Flow[33] with multiplex PCR of the V–J junction of the TCRβ chain (including CDR3 region) followed by sequencing (Supplementary Fig. 1a). TCR sequences retrieved from distinct single sorted cells were compared. Clonotypes were defined either as expanded (i.e. detected in at least two cells) or unique (i.e. detected in no more than one cell). The memory phenotype of individual cells was also recorded during index cell sorting and analyzed post hoc (Supplementary Fig. 1b). Of note, we previously showed that the expression levels of CD45RA, CCR7, and CD27 were minimally affected by PMA/ionomycin stimulation in the presence of Brefeldin A[33].

We analyzed the TCRβ repertoire of single HIV-infected cells in longitudinal blood samples from eight individuals with suppressed plasma viremia on ART for at least two years at the time of the first collection (Supplementary Table 1). There was a median time of 2.2 years (range 1.0–6.5) between the first and the last sample collected. The frequency of p24+ cells measured by HIV-Flow ranged from 0.7 to 1208 cells/$10^6$ CD4+ T cells (Supplementary Fig. 2a), and tended to decrease over time on ART. As previously described[33], p24+ cells preferentially displayed a memory phenotype (Supplementary Fig. 2b) with the central (CD45RA−CD27+CCR7+, $T_{CM}$), transitional (CD45RA-CD27+ CCR7−, $T_{TM}$) and effector (CD45RA−CD27

−CCR7-, $T_{EM}$) memory subsets contributing the most to the pool of infected cells (mean contributions of 12%, 31%, and 46% to the pool of p24+ cells, Supplementary Fig. 2c), whereas p24+ cells were not detected in the naïve subset. Prior to TCRβ sequencing, we also assessed the relative expression of α/β and γ/δ TCR by p24+ cells (Supplementary Fig. 2d) in 5 individuals from our study. None of the p24+ cells expressed a γ/δ TCR, suggesting that our PCR assay optimized for α/β receptors was well-suited to amplify the TCRs from all p24+ cells.

### TCRβ sequencing reveals clonal expansions in the HIV reservoir.

A total of 636 individual TCRβ sequences from single-sorted p24+ cells were retrieved from the 18 samples studied (median of 16 p24+ cells per sample; range 9–194). These individual sequences clustered into 98 different clonotypes, revealing a mix of unique ($n = 69$) and expanded ($n = 29$) TCRβ clonotypes. Duplicated clonotypes were detected in all participants and accounted for the majority of reservoir cells (mean, 74%; range 30–99) (Fig. 1b and Supplementary Fig. 3), confirming the major contribution of clonal expansions to the pool of HIV-infected cells in virally suppressed individuals. In each sample, we observed a median of 2 (range 1–5) independent clonal expansions corresponding to a median of 5 p24+ cells sharing the same TCR (range 2–187). There was no correlation between the contribution of clonal expansions to the pool of p24+ cells and any clinical parameter showed in Supplementary Table 1.

Since infected T cells sharing the same TCR may be the result of clonal expansion or may have been infected by different HIV variants during expansion, we co-amplified the TCR together with the HIV *Env* sequence (C3-V5) in single p24+ cells to distinguish between these two scenarios (Supplementary Fig 4a). TCR and C3-V5 sequences were co-amplified in 10 p24+ cells from one participant. Cells containing duplicated TCRs harbored the exact same viral sequence, which were different than those retrieved in cells harboring distinct TCRs (Supplementary Fig. 4b, c). These results indicated that clonal expansion of an HIV-infected cell is the most likely explanation for the duplication of TCR sequences within the pool of p24+ cells.

### Diversity of the TCRβ repertoire of HIV-infected cells.

To compare the TCR repertoires of HIV-infected and non-infected cells, we applied the same approach to single-sorted p24- cells. As expected, the vast majority (353/357 clonotypes, 99%) of the TCRβ clonotypes retrieved from p24- cells were unique (Fig. 1b and Supplementary Fig. 5). The distribution of V and J segment usage in p24- cells was similar to the human TCR repertoire described in previous studies[34–36], supporting a non-biased TCR amplification (Fig. 2a, b). Interestingly, when excluding the expansion effect by considering each clonotype as unique, the V and J segment usages of distinct TCR clonotypes were similar in p24+ and p24− cells (Fig. 2a, b, respectively), suggesting that the pool of HIV-infected cells was initially established in a large number of T cells with multiple antigen specificity. However, when including duplicated TCRs in the analysis, the V/J combination usage was heavily skewed in the pool of infected cells (Fig. 2c) when compared to p24− control cells (Fig. 2d), suggesting that the bias in the repertoire of the reservoir was attributed to clonal expansions. Altogether, our observations suggest that the restricted TCR diversity observed in the pool of reservoir cells results from antigen-driven clonal expansions.

### Dynamics of clonally expanded HIV-infected cells during ART.

To determine if these clonal expansions of infected cells persisted over time, we performed a longitudinal analysis of the TCR repertoire of p24+ cells in the eight participants. Major clonal

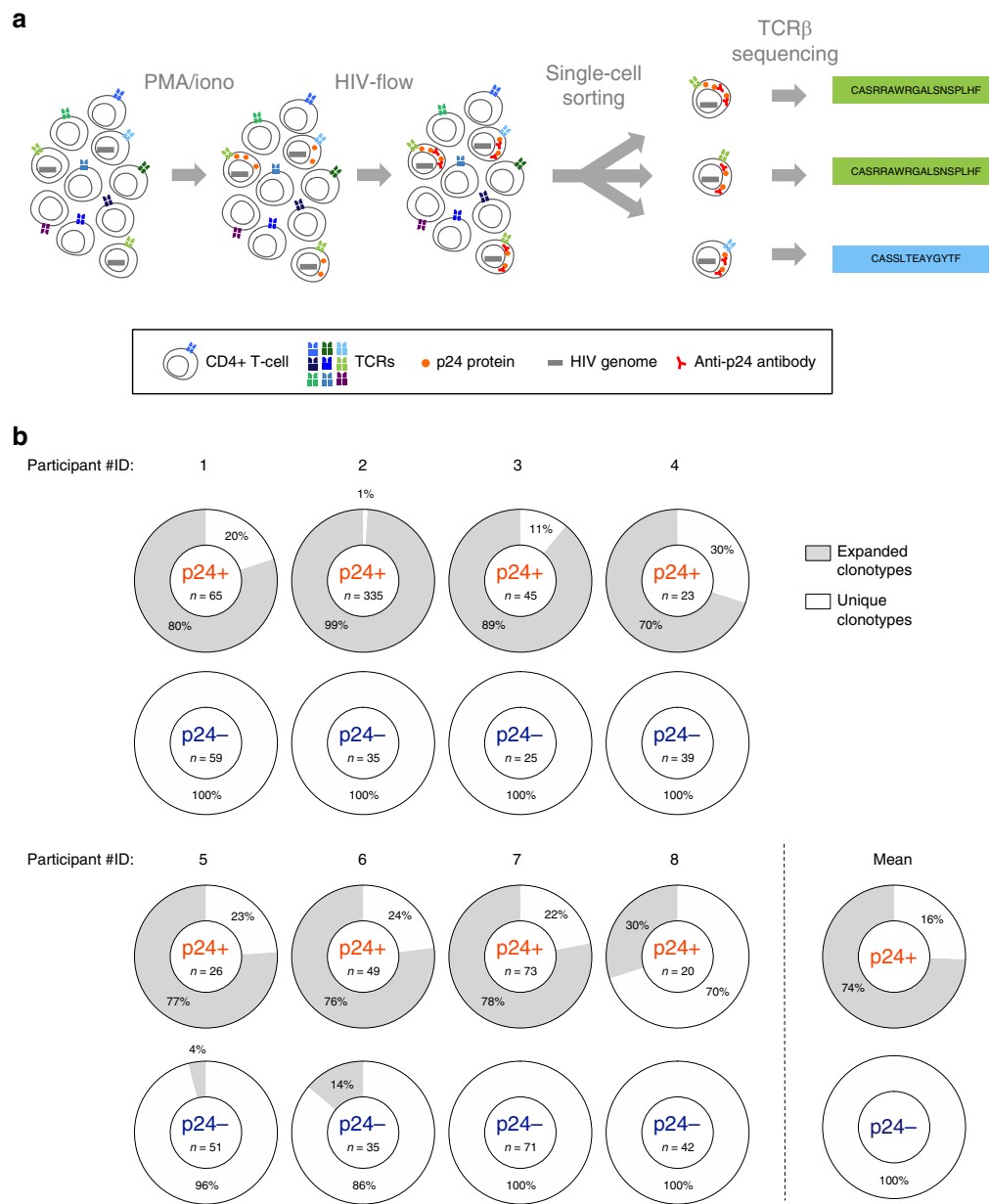

**Fig. 1 Experimental strategy for phenotyping and TCRβ sequencing of single HIV-infected cells. a** Isolated CD4+ T cells were stimulated for 24 h with PMA/ionomycin and single-sorted according to their p24 expression by HIV-Flow. Single sorted p24+ cells underwent a two-round multiplex PCR for amplification of the V–J junction of the TCRβ chain (including CDR3 region) on genomic DNA. CDR3 sequences and V/J-region usage were obtained. **b** The proportion of clonal expansions (in gray) in the pool of p24+ and p24− cells is represented as a pie chart for each participant. Pie charts at the bottom right represent the mean proportion of clonal expansion from all participants. Source data are provided as a Source Data file.

expansions within the reservoir persisted over time in 7/8 participants and for up to 6.5 years (Fig. 3). In addition, transient clonal expansions were observed in several participants (participants #3, #4, #5, #6, and #7) suggesting that clonal expansions followed by contractions were common. Of note, this is likely an underestimate, since the small numbers of cells analyzed may have limited our ability to detect persistent clonal expansions of small magnitude. In some cases, the relative contribution of a given clonotype to the pool of infected cells was maintained over time (participant #1, stability of clonotype 1 over 2.5 years), whereas it varied in others (participant #2, change in the contribution of clonotype 1 versus clonotype 2 over 1.4 years, $p < 0.05$). We conclude that while their proportions may vary over time, infected cellular clonotypes harboring inducible proviruses usually persist during prolonged ART.

**Memory phenotype of clonally expanded HIV-infected cells.** We next sought to determine if the persistence of expanded infected clonotypes was restricted to specific memory CD4+ T cell subsets. Since the memory phenotype of individual p24+ cells was recorded during cell sorting, we were able to analyze the distribution of HIV-infected expanded clonotypes in CD4+ T cells subsets. Expanded p24+ clonotypes were often detected in all three memory subsets ($T_{CM}$, $T_{TM}$, and $T_{EM}$, Fig. 4). All expanded clonotypes systematically displayed at least two different memory phenotypes, which were often maintained over time. As previously reported[28], expanded infected clonotypes were overrepresented in the most differentiated subsets (i.e. $T_{TM}$ and $T_{EM}$). Nonetheless, expanded clonotypes were detected at least in one single-sorted p24+ $T_{CM}$ cell in 7/8 participants (Fig. 4 and Supplementary Fig. 6a–c). Since the developmental differentiation

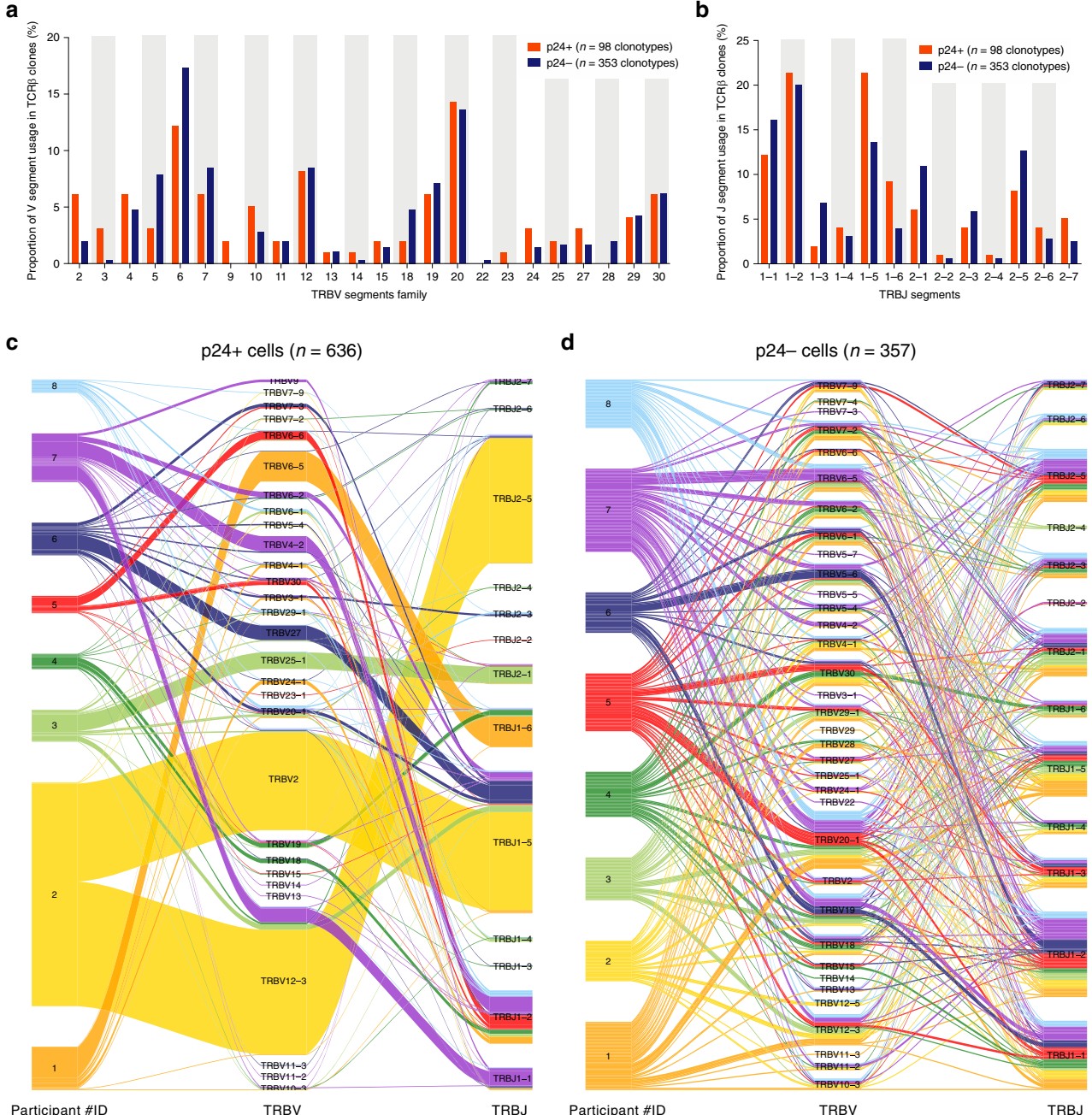

**Fig. 2 The bias in the TCR repertoire of the translation-competent reservoir is due to clonal expansion. a, b** Frequency of TRBV (**a**) and TRBJ (**b**) segment usage for the clonotypes identified by TCRβ sequencing in p24+ cells (red bars, $n = 98$ clonotypes) and p24− cells (blue bars, n = 353 clonotypes). **c, d** Association between the TRBV and TRBJ segments in individual p24 + ($n = 636$, **c**) and p24− ($n = 357$, **d**) cells from all participants (depicted by different colors). Connecting lines between the last two columns represent the number of individual associations. Large connecting lines denote a bias in the TCR repertoire. Source data are provided as a Source Data file.

process of memory CD4+T cells was shown to be linear in the order $T_{CM} > T_{TM} > T_{EM}$[37], this suggested that clonally expanded HIV-infected cells are the progeny of infected $T_{CM}$ cells which proliferated and differentiated into $T_{TM}$ and $T_{EM}$ cells (Supplementary Fig. 6d). In addition, our results indicate that HIV-infected cells have the ability to proliferate and differentiate without being eliminated, suggesting that this process can occur in the absence of HIV production as previously reported[38,39] or that these cells can escape immune-mediated killing, as recently suggested by Ren et al.[40].

**Antigenic specificity of HIV-infected cells.** We next sought to predict the antigenic specificity of the infected cells by comparing the TCR sequence of p24+ clonotypes with public CDR3 sequences. We applied the criteria of Meysman et al.[41] to compare our sequences with those inferred in the McPAS-TCR database[42]. Overall, the frequency of cells with predicted specificity was relatively low both for p24+ ($n = 9/98$ clonotypes) and p24− (14/353 clonotypes) cells (Fig. 5a, b). Among the p24+ cells, some expressed TCR predicted to be reactive to CMV, influenza, *M. tuberculosis* and EBV (Fig. 5c). Interestingly, two of the p24+ clonotypes were expanded. A first expanded clonotype from participant #1 was predicted to be CMV-specific and persisted over time (Fig. 5d), suggesting that persistent antigenic stimulation by CMV may favor the maintenance of HIV-infected cells.

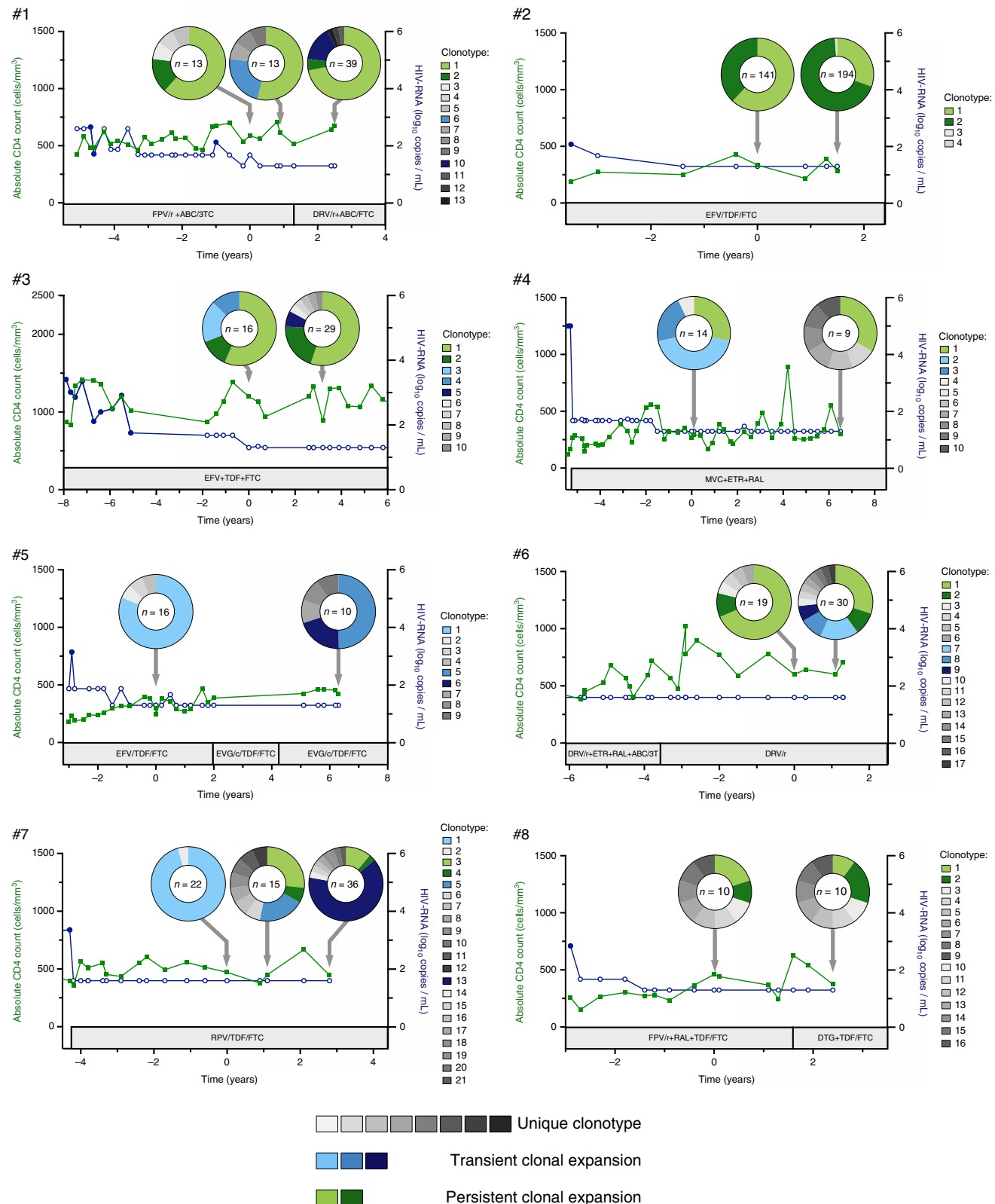

A second clonotype that was predicted to be influenza-specific was largely expanded in the last sample from participant #7 (Fig. 5e), indicating that new and transient antigenic stimulations such as influenza infection or immunization may favor the expansion of influenza-specific HIV-infected cells. Altogether, these results indicate that T cell pools against specific antigens can comprise both infected and uninfected cells and suggest that

reservoir cells from different individuals might be reactive to common antigens. This is in line with the results of recent studies demonstrating that at least a fraction of the HIV reservoir is carried by CMV/EBV and HIV-specific CD4+ T cells[23,43–45].

In summary, our results indicate that antigen-driven clonal expansions highly contribute to the persistence of the translation-competent HIV reservoir in individuals on ART. The phenotypic

**Fig. 3 Dynamics of TCRβ clonotypes in the pool of p24+ cells.** The frequencies of the TCRβ clonotypes in p24+ cells are represented for participants #1 to #8 at each study visit. CD4+ T cell counts (left axis, green lines) and plasma viral loads (right axis, blue lines) are shown. Open blue circles represent undetectable plasma viral load measures and are plotted at the limit of detection of the assay. ART regimens are indicated in the gray boxes. Arrows point the leukapheresis dates. For each sample, the proportion of each clonotype in the pool of p24+ cells is represented in a pie chart. The number of p24+ cells analyzed is indicated in the center of the pie. Expanded clonotypes persisting over time are depicted in shades of green; Expanded clonotypes only detected at a single visit are depicted in shades of blue; Unique clonotypes are depicted in shades of gray. ABC: abacavir; FPV: fosamprenavir; DRV: darunavir; DTG: dolutegravir; EFV: efavirenz; ETR: etravirine; EVG: elvitegravir; FTC: emtricitabine; MVC: maraviroc; /r: ritonavir; RAL: raltegravir; RPV: rilpivirine; TAF: tenofovir alafenamide; TDF: tenofovir disoproxil fumarate; 3TC: lamivudine. Source data are provided as a Source Data file.

analysis suggests that infected T cell clonotypes displaying a differentiated phenotype are the progeny of infected central memory cells undergoing clonal expansion during ART. These findings provide a rationale for the development of therapeutic strategies aimed at limiting antigen-driven proliferation during ART to reduce the pool of infected cells, which may be achieved by decreasing the antigen load of treatable pathogens such as CMV.

## Methods

**Participants and sample collection**. Eight individuals on successful ART were enrolled in this study. All participants underwent longitudinal leukapheresis to collect large numbers of PBMCs. PBMCs were isolated by Ficoll density gradient centrifugation and were cryopreserved in liquid nitrogen.

**Ethics statement**. All participants were adults and signed informed consent forms approved by the McGill University Health Centre, the Centre Hospitalier de l'Université de Montréal and the Martin Memorial Health Systems review boards.

**Antibodies**. p24 KC57-PE was purchased from Beckman Coulter (Cat#6604667, Dilution 1/1000) and p24 28B7-APC was purchased from MediMabs (Cat#MM-0289-APC, Dilution 1/1000). CD3-AF700 (Clone: UCHT-1, Cat#557943, Dilution 1/100), CD45RA-BV786 (Clone: HI100, Cat#563870, Dilution 1/25) and CCR7-BB700 (Clone: 3D12, Cat#566437, Dilution 1/25) were purchased from BD Bioscience. CD8-FITC (Clone: BW135/80, Cat#130-113-719, Dilution 1/100) was purchased from Miltenyi/MACS. CD27-BV421 (Clone: O323, Cat#302823, Dilution 1/50), TCRαβ-FITC (Clone: IP26, Cat#306705, Dilution 1/50) and TCRγδ-PE-Cy7 (Clone: B1 Cat#331221, Dilution 1/50) were purchased from BioLegend. Live/Dead Aqua Cell Stain (405 nm) was purchased from ThermoFisher Scientific (Cat#L34957).

**HIV-Flow procedure**. The HIV-Flow assay was used to quantify and analyze the phenotype of cells expressing p24 protein upon stimulation[33]. Briefly, CD4+ T cells were isolated by negative magnetic selection using the EasySep Human CD4+ T Cell Enrichment Kit (StemCell Technology, Cat#19052). Purity was typically >98%. In all, $5-15 \times 10^6$ CD4+ T cells were resuspended at $2 \times 10^6$ cells/mL in RPMI + 10% Fetal Bovine Serum and antiretroviral drugs were added to the culture medium (200 nM raltegravir, 200 nM lamivudine). Samples were pre-incubated for 1 h with 5 μg/mL Brefeldin A (BFA, Sigma, Cat#B2651) before stimulation in order to prevent the upregulation of cell surface markers, and BFA was maintained in the culture until the end of the stimulation. Cells were then stimulated with 1 μg/mL ionomycin (Sigma, Cat#I9657) and 162 nM PMA (24 h) (Sigma, Cat#P8139). After stimulation, cells were collected, resuspended in PBS and stained with the Aqua Live/Dead staining kit for 30 min at 4 °C. Cells were then stained with antibodies against extracellular molecules in PBS + 4% human serum (Atlanta Biologicals, Cat#540110) for 30 min at 4 °C. After a 45 min fixation/permeabilization step was performed with the FoxP3 Transcription Factor Staining Buffer Set (eBioscience, Cat#00-5523-00) following the manufacturer's instructions, cells were then stained with anti-p24 KC57 and anti-p24 28B7 antibodies for an additional 45 min at RT in the FoxP3 Buffer. Cells were then washed and resuspended in PBS for subsequent cell sorting.

**Flow cytometry cell sorting**. The frequency of p24 double positive cells (KC57+, 28B7+) was determined by flow cytometry in gated viable CD8-CD45RA- T cells. An example of the gating strategy is represented in Supplementary Fig. 7a, b. In all experiments, CD4 + T cells from an HIV-uninfected control were included to set the threshold of positivity. Single p24 double positive (p24+ cells) and double negative cells (p24- cells) were indexed-sorted on a BD FACS ARIA III. Cells were sorted in 96-wells PCR plates containing 7.6 μL of DirectPCR Lysis Reagent (Viagen Biotech) and 0.4 μL of 10 mg/mL proteinase K (from Wisent, 25530–015). The PCR plates were subsequently incubated at 55 °C for 1 h for cell lysis followed by 15 min at 85 °C to inactivate proteinase K. Index-sorting data of p24+ cells were analyzed using FlowJo version 10.5.3.

**TCR amplification on genomic DNA**. We developed a two-step PCR method to amplify a portion of approximately 260 bp of the TCRβ encompassing: (1) the end of the V segment, (2) the CDR3, and (3) the J segment, on genomic DNA from lysed single-cells. We used a set of 22 forward primers complementary to the 23 functional V segments families, and 13 reverse primers complementary to the 13 functional J segments, adapted from Dziubianau et al., to amplify the target portion of the TCRβ in a first multiplex PCR reaction[46]. M13 forward and reverse tags were added to the 5′ end of these primers, to allow a second PCR amplification, which was followed by Sanger sequencing. Sequences of all primers are listed in Supplementary Table 2. The first PCR reaction was performed using the Qiagen Multiplex PCR kit (Qiagen), in a total volume of 50 μL: 25 μL of Qiagen Multiplex PCR master mix, 10 μL of a mix of all primers (each primer at a concentration of 1.25 μM in the mix, providing a final concentration of 250 nM per primer), 5 μL of Q-Solution, and 10 μL of the single-cell lysate. First PCR conditions were as follows: 15 min at 95 °C followed by 40 cycles of; 30 s at 95 °C, 90 s at 68 °C and 20 s at 72 °C; with a final elongation for 5 min at 72 °C. A second round of PCR reaction was performed using the M13F and M13R primers (see Supplementary Table 2) and the Taq DNA Polymerase kit (Invitrogen), in a total volume of 50 μL: 5 μL of 10x PCR buffer, 3 μL of $MgCl_2$ (50 mM), 1.5 μL of dNTPs (10 mM), 2 μL of M13F primer and 2 μL of M13R primer (each at 10 μM, providing a final concentration of 400 nM per primer), 0.5 μl Taq DNA Polymerase (5 U/μL), 26 μL $H_2O$ and 10 μL of the first PCR products. The amplification conditions for the second PCR reaction were as follows: 15 min at 95 °C followed by 40 cycles of; 30 s at 95 °C, 90 s at 57 °C and 30 s at 72 °C; with a final elongation of 5 min at 72 °C. When co-amplifying TCR and *Env* C3-V5 sequences, *Env* primers were added to the first PCR reaction, under the same amplification conditions. The second PCRs were performed separately for TCR and *Env*, using the same amplification conditions (see *Env* primers in Supplementary Table 2).

**TCR sequencing and analysis**. Successful amplification of the TCRβ region was verified by electrophoresis on a 2% agarose gel and followed by gel purification of the TCRβ bands using the Buffer QG and the QIAquick 96 PCR Purification kit (Qiagen), according to the manufacturer's instructions. Sanger sequencing was performed by Eurofins Genomics, with M13F and M13R as sequencing primers. TCRβ sequences were re-constructed using both forward and reverse sequences, and were analyzed using the V-QUEST tool of the IMGT® database (IMGT®, the international ImMunoGeneTics information system®, http://www.imgt.org[47]) to retrieve TCRβ information, including V and J segments usage and junction/CDR3 analysis (example in Supplementary Fig. 1a). TCR sequences were analyzed using an algorithm to predict antigen specificity: CDR3 sequences were compared to the McPAS-TCR database of TCRs of known antigenic specificity (http://friedmanlab.weizmann.ac.il/McPAS-TCR/[42]) and sequence similarities were identified. We predicted TCR specificity using the three criteria described by Meysman et al.[41]: (1) CDR3 sequences should have identical length, (2) CDR3 sequences should be long enough and (3) CDR3 sequences should not differ by more than one amino acid. Among all CDR3 sequences, those fulfilling these three criteria with matched CDR3 sequences from the database were considered at high probability of sharing the same specificity.

**Data representations and statistical analyzes**. A chord diagram displaying inter-relationships between TCR clonotypes and the memory phenotype was plotted for each sample using the program of Circular Visualization within the circlize package (version 0.4.8)[48] in R version 3.1.1 (R Foundation, Vienna, Austria). Sankey diagrams representing the V/J associations per participant for p24+ and p24- cells were generated using the alluvial (version 0.1-2) package in R.

All other data were analyzed and represented using Graphpad Prism v6.0 h. Results were represented as median or mean values, with interquartile range or minimum and maximum values, as indicated in the figure legends. Correlations were determined using nonparametric Spearman's test. For group comparisons, non-parametric Wilcoxon matched-pairs signed rank tests were used. P values of less or equal to 0.05 were considered statistically significant.

**Statistics and reproducibility**. To ensure the reproducibility of our approach, one sample (participant ID#1, visit 3) was repeated three times. A similar distribution of the T cell clonotypes within the reservoir, was detected in each independent

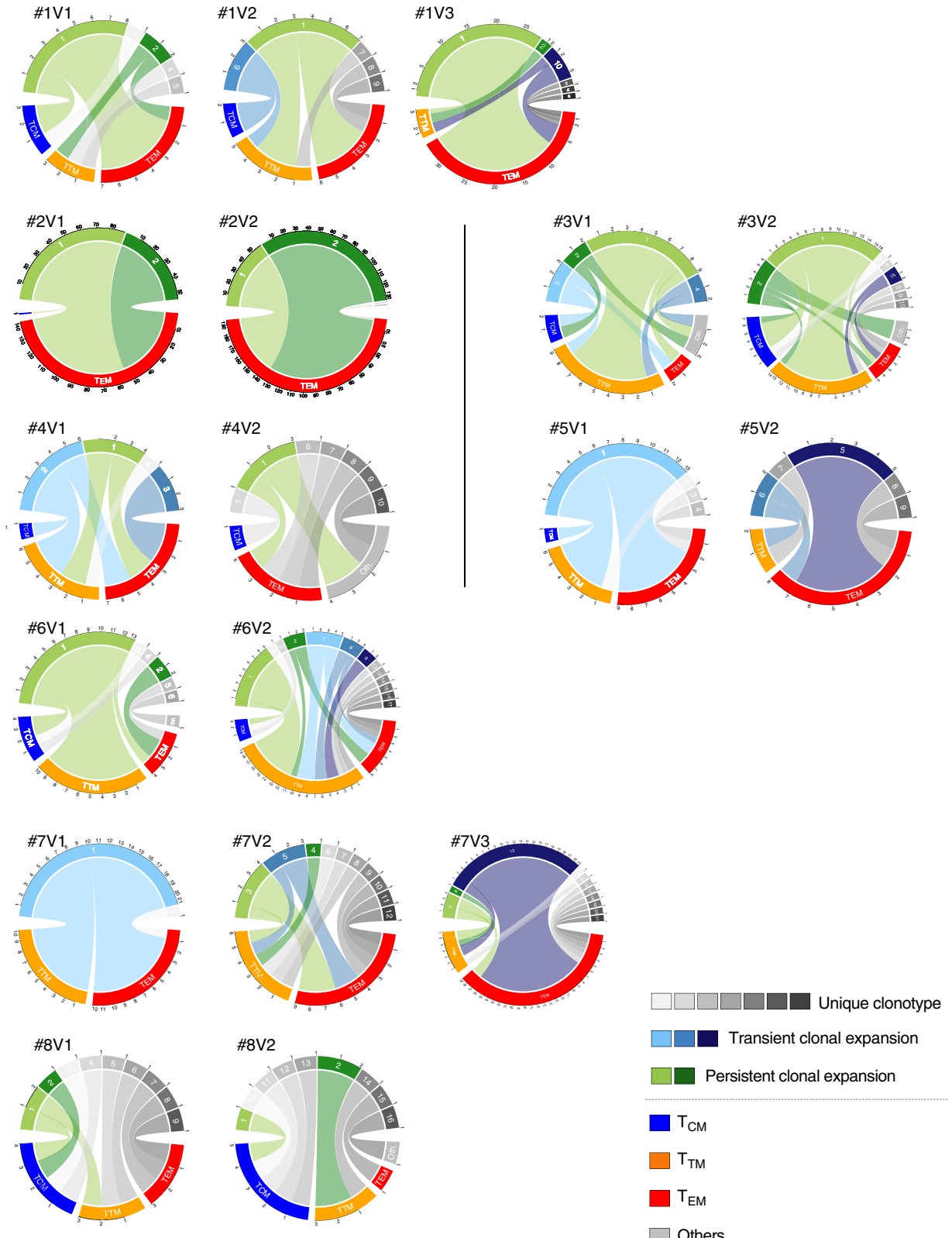

**Fig. 4 TCRβ clonotypes from p24+ cells display multiple memory phenotypes.** The distribution of TCRβ clonotypes of p24+ cells among memory subsets is represented as a chord diagram for participants #1 to #8 at each study visit (V1, V2, and V3). The circular representation shows the link between a specific clonotype (top half of the circle) and its memory phenotype (bottom half of the circle). The circular axis represents the number of p24+ cells in each clonotype/subset. Expanded clonotypes persisting over time are depicted in shades of green; Expanded clonotypes detected at a single visit are depicted in shades of blue; Unique clonotypes are depicted in shades of gray. The memory subset color code is as follows: central memory ($T_{CM}$) in blue; transitional memory ($T_{TM}$) in orange; effector memory ($T_{EM}$) in red; and undefined (others) in gray. Source data are provided as a Source Data file.

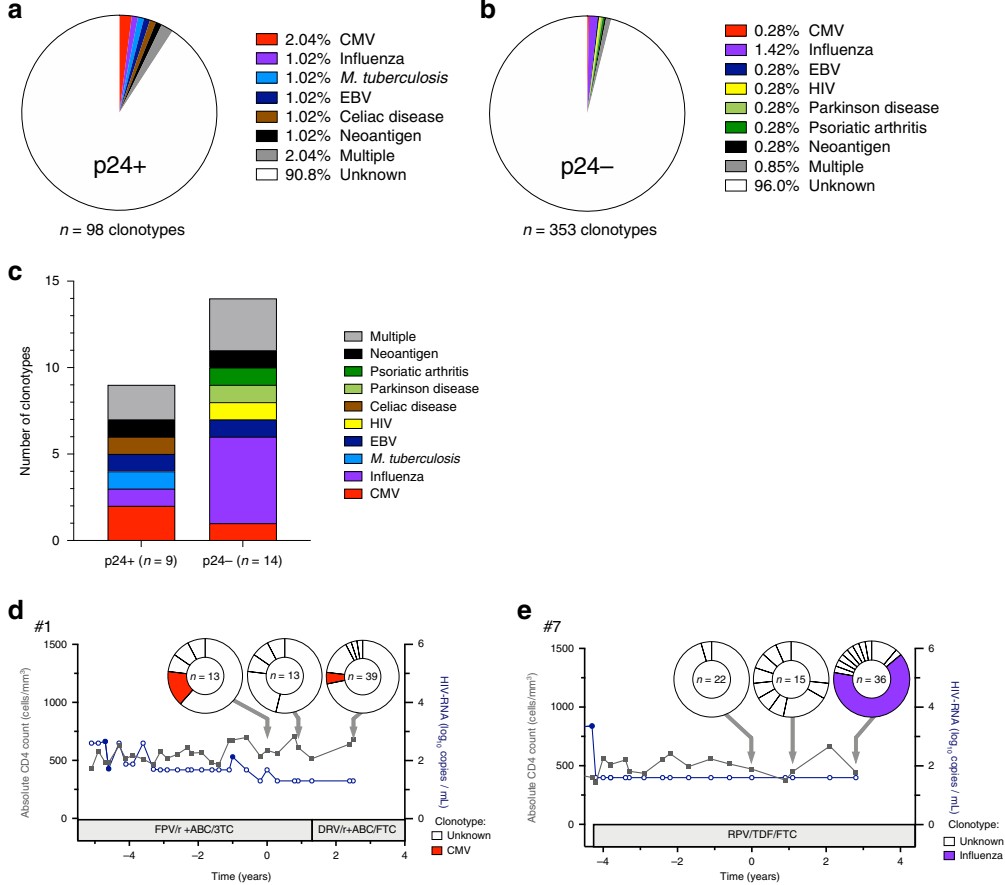

**Fig. 5 Predicted antigen specificity of p24+ cells. a**, **b** Pie charts depicting the proportion of clonotypes with predicted antigen specificity in p24+ (**a**) and p24− (**b**) cells. **c** Number of p24+ (n = 9) and p24− (n = 14) clonotypes for which the antigen specificity could be identified by TCR similarity analysis. **d**, **e** Two clonotypes with identified specificity were expanded. **d** A CMV-specific expanded clonotype was identified in participant #1 and persisted over time on ART. **e** An influenza-specific expanded clonotype was identified in the third sample of participant #7. Source data are provided as a Source Data file.

experiment. The results from the three experiments were combined to generate Fig. 3.

**Reporting summary**. Further information on research design is available in the Nature Research Reporting Summary linked to this article.

## Data availability

All data generated or analyzed during this study are included in this published article and its Supplementary Information files. The source data underlying both the main and Supplementary Figs. are provided as a Source Data file.

External databases used in this study are available online: IMGT® database (IMGT®, the international ImMunoGeneTics information system®, http://www.imgt.org); McPAS-TCR database (http://friedmanlab.weizmann.ac.il/McPAS-TCR/).

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

## Acknowledgements

The study team is grateful to the individuals who volunteered to participate in this study. We thank Josée Girouard, Mario Legault and Brenda Jacobs for recruitment, coordination and clinical assistance with study participants. We thank the flow cytometry core at the CRCHUM, managed by Dominique Gauchat and Philippe St-Onge for cell sorting as well as the NC3 core (Olfa Debbeche). We also thank Nicole Bernard and Tsoarello Mabanga for HLA-typing. The authors thank Steven G Deeks, Peter Hunt and Mark Brockman for advice and helpful discussions. This work was supported by the Canadian Institutes for Health Research (CIHR; operating grant #364408 and the Canadian HIV Cure Enterprise (CanCURE) Team Grant HB2 - 164064), the National Institute of Allergy and Infectious Diseases, National Institute for Drug Abuse, National Institute of Mental Health, and the National Institute of Neurological Disorders and Stroke [NIAID/NIDA/NIMH/NINDS, grant number UM1AI126611: Delaney AIDS Research Enterprise (DARE) to Find a Cure], the Foundation for AIDS Research (amfAR, Research Consortium on HIV Eradication 108687-54-RGRL and 108928-56-RGRL), the Réseau SIDA et maladies infectieuses du Fonds de Recherche du Québec - Santé (FRQ-S). P.G. is supported by a postdoctoral fellowship from CIHR (#415209). J.P.R. is the holder of the Louis Lowenstein Chair in Hematology and Oncology, McGill University. N.C. is supported by Research Scholar Career Awards of the FRQ-S (#253292). The funders had no role in study design, data collection and analysis, decision to publish, or preparation of the manuscript.

## Author contributions

P.G. and N.C. wrote the paper; P.G., M.P., R.F., and N.C. designed and analyzed experiments; P.G., A.P. performed the HIV-Flow, cell sorting and TCR sequencing experiments. M.R. and J.P.R., performed study subject recruitment and oversaw sample collection. All authors read and edited the paper.

## Competing interests

The authors declare no competing interests.
