## [Peer Review File · Nature Communications]

Reviewers' Comments:

Reviewer #1:

Remarks to the Author:

In this manuscript the authors explored the dynamics of the inducible HIV reservoir in longitudinal blood samples from 8 HIV-infected patients. Under the assumption that enrichment in specific clonotypes would be the result of clonal expansion, they coupled flow cytometry-assisted sorting of p24+ induced cells with sequencing of TRB. The authors observed a bias in the induced reservoir repertoire attributed to clonal expansions and hypothesized that infected clones with a highly differentiated phenotype are the result of central memory cells undergoing clonal expansion.

While the conclusions are mostly confirmative, the use of TCR sequencing to follow the reservoir dynamic is the strength of the paper and an original point worth mentioning. Overall the paper is clear and well written. Found below a few comments.

- For some of the patient involved in this work (#4 , #5 and #8 namely) sorted p24+ cells are very few, which can limit the meaningfulness of the data. Therefore, statements such as "In addition, transient clonal expansions followed by contractions were common (participants #4, #5 and #7" (line 113) are highly speculative.
- In this paper the authors used 5-15.106 CD4 T cells for stimulation/staining prior to sorting. Given the size of the reservoir, is that enough to avoid bias due to sampling effect?
- To reinforce the authors' conclusions, it would have been interesting to associate their approach with some measures of the reservoir within clonal expansions of CD4 T cells specific for antigens from viruses like CMV or EBV.
- Supplemental Table 2 contains 19 samples since it also includes a third visit for patient #3 that has not been analyzed (only 3 p24+ cells). The authors should remove this sample to avoid confusion, especially since the main text refers to 18 samples (line 84).

Reviewer #2:

Remarks to the Author:

Gantner and colleagues have performed experiments aimed at understanding the clonality of the HIV-1 latent reservoir. They use a recently developed method to identify p24+ HIV-1 infected cells by flow cytometry and then perform sophisticated molecular biology to recover TCR sequence from lysed, single, sorted cells. This process allowed them to recover and compare TCRs from p24+ HIV infected cells to those in p24 negative, HIV uninfected cells from the same donors. These experiments demonstrated that infected cells which are induced by stimulation in vitro have a higher degree of TCR clonality than uninfected cells treated in the same way. The manuscript here presents data which advances the field of HIV-1 reservoir biology. The authors should be commended for their novel approach and depth of informative data. The findings presented here will extend beyond the HIV-1 community and will be important for anyone studying persistent pathogens and/or the human immune response.

However, there are a number of issues whose resolution would strengthen this manuscript. Please see below for a list of major and minor points:

Major:

1. Virus sequencing or viral integration site sequencing is required to conclusively demonstrate the author's claims that these specific T cells' proliferation maintains the latent reservoir. As the data stands now, the authors have demonstrated that clones of T cells are infected – however, the data cannot distinguish between the scenario where a large clone of infected cells at one site were all simultaneously infected by different viruses and are maintained in vivo independently of the specific virus or virus integration site versus a single T cell becoming infected and carrying its provirus through successive rounds of proliferation. Thus, it is important to prove that these cells are part of the same viral clone and T cell clone. Because the authors have shown the ability to amplify genomic regions from lysed single sorted cells, they should also be able to use methods for amplification of the entire genome to pull out either virus sequence or integration site information.
2. One of the major conclusions of this manuscript involves the characterization of memory T cell subset. However, the authors treat cells for 24h with PMA and ionomycin to induce expression of p24. The use of Brefeldin A to prevent the upregulation of cell surface markers was used. However, it's my understanding that BFA doesn't prevent downregulation of all markers – thus it will be important to show that BFA doesn't dramatically alter phenotyping of cells, live cell frequency, or the overall frequency of p24+ cells.

Minor:

1. Supplemental Figure 1 is a little confusing – defining "clone" is important here. Are clones 3, 4, and 5 cells that were only identified once? Relatedly, please define how clones were identified. It is unclear from the methods how the authors classified something as an expanded clone.
2. Lines 101-104 are unclear, but seem like an important point. Consider revising.
3. The authors should consider citing Brad Jones' recent paper showing that productively infected cells escape CTL killing during the discussion of how HIV infected cells can escape killing on line 137.
4. The discussion of reservoir cells from different individuals being reactive to common antigens should consider citing the early paper from Tony Fauci's group looking at influenza immunization and HIV expression, a paper from Tim Henrich in 2017 looking at EBV/CMV specific cells during T cell reconstitution after cell-ablative chemotherapy, a paper from Hey-Nguyen and colleagues looking at the antigen specificity of HIV-infected CD4+ cells, and the recent preprint from Mendoza and colleagues looking at the antigen specificity of the HIV proviral reservoir.
5. Participant #2 in Supplemental Table 1 seems to have more copies of integrated DNA than total DNA – please double check this data.
6. Figure 2c and d are difficult to interpret. Is there a simpler way to display this data?

Reviewer #1:

In this manuscript the authors explored the dynamics of the inducible HIV reservoir in longitudinal blood samples from 8 HIV-infected patients. Under the assumption that enrichment in specific clonotypes would be the result of clonal expansion, they coupled flow cytometry-assisted sorting of p24+ induced cells with sequencing of TRB. The authors observed a bias in the induced reservoir repertoire attributed to clonal expansions and hypothesized that infected clones with a highly differentiated phenotype are the result of central memory cells undergoing clonal expansion.

While the conclusions are mostly confirmative, the use of TCR sequencing to follow the reservoir dynamic is the strength of the paper and an original point worth mentioning. Overall the paper is clear and well written. Found below a few comments.

- For some of the patients involved in this work (#4, #5 and #8 namely) sorted p24+ cells are very few, which can limit the meaningfulness of the data. Therefore, statements such as “In addition, transient clonal expansions followed by contractions were common (participants #4, #5 and #7)” (line 113) are highly speculative.

We agree that for some of the samples included in our study, the number of cells analyzed was limited (around 10 p24+ cells). Of note, even with these low numbers of cells which could have limited our ability to detect the same TCR sequence twice, we observed duplicated TCRs in all samples. This indicates that clonal expansions are actually quite common and represent a large proportion of infected cells in all our participants. We agree that this limited number of clones may have limited our ability to detect persistent expansion over time, particularly if these were present at low frequencies. To address the reviewer’s concern, we modified our statement as follows (lines 128-132):

“In addition, transient clonal expansions were observed in several participants (participants #3, #4, #5, #6 and #7) suggesting that clonal expansions followed by contractions were common. Of note, this is likely an underestimate since the small numbers of cells analyzed may have limited our ability to detect persistent clonal expansions of small magnitude”.

- In this paper the authors used 5-15.106 CD4 T cells for stimulation/staining prior to sorting. Given the size of the reservoir, is that enough to avoid bias due to sampling effect?

We thank the reviewer for his comment. We acknowledge that the sampling effect may be an issue when sampling small numbers of cells. Our approach allows to identify major clonal expansions in the reservoir, and it is likely that small clonal expansions were not captured. We would like to emphasize that even with a small number of cells, the repertoire of p24+ cells was highly reproducible in independent experiments. For the reviewer’s consideration, we show below the results of three experiments performed on the same sample. Strikingly, we observed a similar distribution of the T cell clones within the reservoir, with clonotype #1 (major in green) and #10 (minor in blue) being detected

in each independent experiment. We believe this demonstrates the reproducibility of our approach.

- To reinforce the authors' conclusions, it would have been interesting to associate their approach with some measures of the reservoir within clonal expansions of CD4 T cells specific for antigens from viruses like CMV or EBV.

We thank the reviewer for this interesting suggestion. Recently, Mendoza *et al.* (1) measured responses of persistently infected cells to a small subset of antigens from viruses that produce chronic or recurrent infections (CMV, EBV, influenza). They identified clones of antigen-responsive CD4+ T cells containing defective and intact latent proviruses in the majority of the individuals studied. We included this reference in our revised manuscript.

Although we believe that our single-cell approach may provide a more quantitative assessment of antigen driven clonal expansion in the reservoir, our initial analysis identified only a few antigen specificities. To address the reviewers' comment and to reinforce our conclusions regarding the importance of antigen-driven clonal expansions in the persistence of the reservoir, we re-analyzed our dataset using a different approach. Several algorithms have been developed to predict antigen specificity (2-4) from TCR sequences. These approaches generally use TCR β chain sequences and compare them with public databases looking for sequence similarities. Recently, Meysman *et al.* (5) compared several algorithms to predict TCR specificity based on sequence similarities. They conclude that "one can achieve reasonable clustering of epitope-specific TCR sequences based on three simple criteria: 1) if they have identical length, 2) if the CDR3 amino sequence is sufficiently long and 3) if they differ by at most one amino acid". We applied these three criteria to assess the specificity of our TCR β sequences based on the similarities of TCR sequences inferred in the McPAS-TCR database (6). This database gathers more than 21,000 TCR sequences with their associated antigen as well as details about the source information.

Overall, we identified putative specificities for 9 p24+ and 14 p24- clonotypes. Among the p24+ cells, some expressed TCR predicted to be reactive to CMV, influenza, *M. tuberculosis* and EBV. Interestingly, two of these clonotypes were expanded. A first expanded clonotype from participant #1 was predicted to be CMV-specific and persisted over time, suggesting that persistent antigenic stimulation such as CMV can favor the

maintenance of HIV-infected cells. A second clonotype, which was predicted to be influenza-specific, was largely expanded in the last sample from participant #7, indicating that new and transient antigenic stimulations such as influenza infections or immunizations may have contributed to the expansion of influenza-specific HIV-infected cells.

We believe that this new analysis reinforces our initial conclusions. We included these results as a new main Figure (Figure 5), as a replacement of our initial cluster analysis and revised the manuscript accordingly (lines 158-175).

- Supplemental Table 2 contains 19 samples since it also includes a third visit for patient #3 that has not been analyzed (only 3 p24+ cells). The authors should remove this sample to avoid confusion, especially since the main text refers to 18 samples (line 84).

We apologize for submitting the incorrect version of Supplementary Table 2. We originally excluded data from the third visit of participant #3 because only 3 p24+ cells were sorted. We have corrected the table in the revised manuscript.

Reviewer #3:

Gantner and colleagues have performed experiments aimed at understanding the clonality of the HIV-1 latent reservoir. They use a recently developed method to identify p24+ HIV-1 infected cells by flow cytometry and then perform sophisticated molecular biology to recover TCR sequence from lysed, single, sorted cells. This process allowed them to recover and compare TCRs from p24+ HIV infected cells to those in p24 negative, HIV uninfected cells from the same donors. These experiments demonstrated that infected cells which are induced by stimulation *in vitro* have a higher degree of TCR clonality than uninfected cells treated in the same way. The manuscript here presents data which advances the field of HIV-1 reservoir biology. The authors should be commended for their novel approach and depth of informative data. The findings presented here will extend beyond the HIV-1 community and will be important for anyone studying persistent pathogens and/or the human immune response.

However, there are a number of issues whose resolution would strengthen this manuscript. Please see below for a list of major and minor points:

Major:

1. Virus sequencing or viral integration site sequencing is required to conclusively demonstrate the author's claims that these specific T cells' proliferation maintains the latent reservoir. As the data stands now, the authors have demonstrated that clones of T cells are infected – however, the data cannot distinguish between the scenario where a large clone of infected cells at one site were all simultaneously infected by different viruses and are maintained *in vivo* independently of the specific virus or virus integration site versus a single T cell becoming infected and carrying its provirus through successive rounds of proliferation. Thus, it is important to prove that these cells are part of the same viral clone and T cell clone. Because the authors have shown the ability to amplify genomic regions from lysed single sorted cells, they should also be able to use methods for amplification of the entire genome to pull out either virus sequence or integration site information.

The reviewer raised an important point and we agree that two scenarios may explain our observations: (1) an expanded T cell clonotype could be infected by several HIV variants during or after T cell expansion, which would result in a relatively diverse proviral landscape within a given clonotype; or (2) an individual T cell could be infected by a single HIV variant and then expand, possibly as a result of antigenic stimulation, which would result in a single viral variant within a given expanded clonotype. Since most studies analyzing integration sites and near full-length HIV sequences reported that duplicated sequences are common within the reservoir, the second scenario is likely to be predominant *in vivo*. Moreover, Cohn *et al.* directly addressed this question by investigating both TCR and viral RNA sequences in single infected cells using the combination of the LURE assay and RNAseq (7). The analysis of samples from three different donors, showed that within a clonal expansion, all viral sequences were identical. These data are in favor of the second scenario: one clonal expansion (TCR) matches with one viral sequence.

To more directly address the reviewer's comment, we modified our TCR amplification assay by adding a pair of primers specific of a highly variable region of the HIV envelope (C3-V5). We focused on a small portion of the HIV genome and not the entire HIV genome because the size of the TCR amplicons (approximately 260bp) and the HIV genome (approximately 10,000bp) are too different to be combined in a single amplification. By amplifying simultaneously both TCR β chain and HIV C3-V5 *env* sequences in single-sorted p24⁺ cells, we sought to verify if these sequences matched. TCR and C3-V5 sequences were co-amplified in 10 p24⁺ cells from one participant. Based on the TCR sequence, there were 2 clonal expansions (n=2 and n=3 cells, respectively) and 5 unique clonotypes. Cells containing duplicated TCRs harbored the exact same viral sequence, which was different from the other cells. This result strongly suggests that in this participant, an expanded TCR clonotype was mostly infected by a single HIV variant.

We added these data in the result section and in supplementary Figure 4 and present the two scenarios suggested by the reviewer in the revised manuscript (lines 101-109).

2. One of the major conclusions of this manuscript involves the characterization of memory T cell subset. However, the authors treat cells for 24h with PMA and ionomycin to induce expression of p24. The use of Brefeldin A to prevent the upregulation of cell surface markers was used. However, it's my understanding that BFA doesn't prevent downregulation of all markers – thus it will be important to show that BFA doesn't dramatically alter phenotyping of cells, live cell frequency, or the overall frequency of p24⁺ cells.

We thank the reviewer for this important comment. The memory phenotype following stimulation with PMA/ionomycin in the presence of BFA was analyzed in our previous manuscript describing the HIV-Flow assay (8). As shown below (supplementary Figure 8, panel A in Pardons *et al.*), although some changes were noted in the shape of the subsets, the frequencies of the different memory subsets were not significantly altered by the stimulation in the presence of BFA.

We added the following sentence to the revised manuscript (lines 70-72):

“Of note, we previously showed that the expression levels of CD45RA, CCR7 and CD27 were minimally affected by PMA/ionomycin stimulation in the presence of Brefeldin A (8).”

Minor:

1. Supplemental Figure 1 is a little confusing – defining “clone” is important here. Are clones 3, 4, and 5 cells that were only identified once? Relatedly, please define how clones were identified. It is unclear from the methods how the authors classified something as an expanded clone.

We apologize for the lack of clarity. We define a clonotype as a T cell expressing a given TCR sequence. Clonotypes can be expanded (i.e. detected in at least 2 cells) or unique (i.e. detected in no more than 1 cell). We clarified this in the revised manuscript by using only “clonotype” and by adding the following definition (lines 66-69):

“TCR sequences retrieved from distinct single sorted cells were compared. Clonotypes were defined either as expanded (i.e. detected in at least two cells) or unique (i.e. detected in no more than one cell).”

2. Lines 101-104 are unclear, but seem like an important point. Consider revising.

Here we analyzed the TCR diversity without taking into account the duplication (expansion) of some clonotypes. By considering all TCRs as unique, the V and J associations were similar between p24+ and p24-, suggesting that the initial infection took place in a non-biased TCR repertoire (representative of the total population of cells), that further became biased as a result of clonal expansion.

We rephrased the paragraph as follows (lines 115-119):

“Interestingly, when excluding the expansion effect by considering each clonotype as unique, the V and J segment usages of distinct TCR clonotypes were similar in p24+ and p24- cells (Fig. 2a and 2b, respectively), suggesting that the pool of HIV-infected cells was initially established in a large number of T cells with multiple antigen specificity.”

3. The authors should consider citing Brad Jones’ recent paper showing that productively infected cells escape CTL killing during the discussion of how HIV infected cells can escape killing on line 137.

We thank the reviewer for this suggestion and quoted the recent paper from Brad Jones and colleagues.

4. The discussion of reservoir cells from different individuals being reactive to common antigens should consider citing the early paper from Tony Fauci’s group looking at influenza immunization and HIV expression, a paper from Tim Henrich in 2017 looking at EBV/CMV specific cells during T cell reconstitution after cell-ablative chemotherapy, a paper from Hey-Nguyen and colleagues looking at the antigen specificity of HIV-infected CD4+ cells, and the recent preprint from Mendoza and colleagues looking at the antigen specificity of the HIV proviral reservoir.

We appreciate the reviewer’s suggestions and apologize for not including these references in our original manuscript. We extensively modified this part of the manuscript (see our response to Reviewer #1) and included all the references suggested by the Reviewer (lines 158-171)

5. Participant #2 in Supplemental Table 1 seems to have more copies of integrated DNA than total DNA – please double check this data.

We double checked these values and confirm that in this participant, the integrated HIV DNA levels exceeded total HIV DNA levels. Among all the samples we analyzed during the past 10 years using these assays, higher levels of integrated HIV DNA than total HIV DNA were rarely observed (<3%). A likely explanation for this is the deletion of a portion of the *gag* gene in a large proportion of proviruses in these participants since the total assay uses two HIV primers (LTR and *gag*), whereas the integrated assay uses a single HIV primer (LTR) together with *alu* primers.

6. Figure 2c and d are difficult to interpret. Is there a simpler way to display this data?

The objective of these figures was to show that the TCR repertoire of the p24+ cells is heavily biased compared to the p24- negative control, which is highlighted by the “thick” links between V and J segments in Figure 2c. This type of representation has the advantage of showing the association between V and J segments and to take into account the number of cells analyzed. We would prefer to keep the figure as is but we will be happy to modify it if the reviewer feels strongly about it.

References:

1. Mendoza P, Jackson JR, Oliveira TY, Gaebler C, Ramos V, Caskey M, Jankovic M, Nussenzweig MC, Cohn LB. 2020. Antigen-responsive CD4+ T cell clones contribute to the HIV-1 latent reservoir. *J Exp Med* 217.
2. Glanville J, Huang H, Nau A, Hatton O, Wagar LE, Rubelt F, Ji X, Han A, Krams SM, Pettus C, Haas N, Arlehamn CSL, Sette A, Boyd SD, Scriba TJ, Martinez OM, Davis MM. 2017. Identifying specificity groups in the T cell receptor repertoire. *Nature* 547:94-98.
3. Dash P, Fiore-Gartland AJ, Hertz T, Wang GC, Sharma S, Souquette A, Crawford JC, Clemens EB, Nguyen THO, Kedzierska K, La Gruta NL, Bradley P, Thomas PG. 2017. Quantifiable predictive features define epitope-specific T cell receptor repertoires. *Nature* 547:89-93.
4. Thakkar N, Bailey-Kellogg C. 2019. Balancing sensitivity and specificity in distinguishing TCR groups by CDR sequence similarity. *BMC Bioinformatics* 20:241.
5. Meysman P, De Neuter N, Gielis S, Bui Thi D, Ogunjimi B, Laukens K. 2019. On the viability of unsupervised T-cell receptor sequence clustering for epitope preference. *Bioinformatics* 35:1461-1468.
6. Tickotsky N, Sagiv T, Prilusky J, Shifrut E, Friedman N. 2017. McPAS-TCR: a manually curated catalogue of pathology-associated T cell receptor sequences. *Bioinformatics* 33:2924-2929.
7. Cohn LB, da Silva IT, Valieris R, Huang AS, Lorenzi JCC, Cohen YZ, Pai JA, Butler AL, Caskey M, Jankovic M, Nussenzweig MC. 2018. Clonal CD4(+) T cells in the HIV-1 latent reservoir display a distinct gene profile upon reactivation. *Nat Med* 24:604-609.
8. Pardons M, Baxter AE, Massanella M, Pagliuzza A, Fromentin R, Dufour C, Leyre L, Routy JP, Kaufmann DE, Chomont N. 2019. Single-cell characterization and quantification of translation-competent viral reservoirs in treated and untreated HIV infection. *PLoS Pathog* 15:e1007619.
9. Henrich TJ, Hobbs KS, Hanhauser E, Scully E, Hogan LE, Robles YP, Leadabrand KS, Marty FM, Palmer CD, Jost S, Korner C, Li JZ, Gandhi RT, Hamdan A, Abramson J, LaCasce AS, Kuritzkes DR. 2017. Human Immunodeficiency Virus Type 1 Persistence Following Systemic Chemotherapy for Malignancy. *J Infect Dis* 216:254-262.
10. Demoustier A, Gubler B, Lambotte O, de Goer MG, Wallon C, Goujard C, Delfraissy JF, Taoufik Y. 2002. In patients on prolonged HAART, a significant pool of HIV infected CD4 T cells are HIV-specific. *AIDS* 16:1749-54.
11. Hey-Nguyen WJ, Bailey M, Xu Y, Suzuki K, Van Bockel D, Finlayson R, Leigh Brown A, Carr A, Cooper DA, Kelleher AD, Koelsch KK, Zaunders JJ. 2019. HIV-1 DNA Is Maintained in Antigen-Specific CD4+ T Cell Subsets in Patients on Long-Term Antiretroviral Therapy Regardless of Recurrent Antigen Exposure. *AIDS Res Hum Retroviruses* 35:112-120.

Reviewers' Comments:

Reviewer #1:

Remarks to the Author:

The authors have provided satisfactory replies to my comments

Reviewer #2:

Remarks to the Author:

The authors have addressed all my concerns and comments thoroughly and I would like to commend them for a beautiful piece of work.